# Mixed Neuroendocrine Non-Neuroendocrine Neoplasms of the Gastrointestinal Tract: A Case Series

**DOI:** 10.3390/healthcare10040708

**Published:** 2022-04-11

**Authors:** Luigi Pio Guerrera, Gabriella Suarato, Rossella Napolitano, Alessandra Perrone, Vincenza Caputo, Anna Ventriglia, Giulia Martini, Carminia Maria Della Corte, Michele Orditura, Erika Martinelli, Fortunato Ciardiello, Marco Montella, Renato Franco, Teresa Troiani, Stefania Napolitano

**Affiliations:** 1Medical Oncology, Department of Precision Medicine, Università degli Studi della Campania “Luigi Vanvitelli”, 80131 Napoli, Italy; luigipioguerrera@hotmail.it (L.P.G.); gabriella.suarato@gmail.com (G.S.); rossella.napolitano@unicampania.it (R.N.); alessandra.perrone@unicampania.it (A.P.); vincenza.caputo.92@gmail.com (V.C.); anna.ventriglia@gmail.com (A.V.); giulia.martini@unicampania.it (G.M.); carminiamaria.dellacorte@unicampania.it (C.M.D.C.); michele.orditura@unicampania.it (M.O.); erika.martinelli@unicampania.it (E.M.); fortunato.ciardiello@unicampania.it (F.C.); teresa.troiani@unicampania.it (T.T.); 2Pathology Unit, Department of Mental and Physical Health and Preventive Medicine, Università degli Studi della Campania “Luigi Vanvitelli”, 80138 Napoli, Italy; marco.montella@unicampania.it (M.M.); renato.franco@unicampania.it (R.F.)

**Keywords:** MiNENs, mixed neuroendocrine/non-neuroendocrine neoplasms, gastrointestinal tumors, NGS analysis, Foundation One (F1)

## Abstract

Mixed neuroendocrine non-neuroendocrine neoplasms (MiNENs) refer to heterogenous rare neoplasms constituted of at least a neuroendocrine population—either well-differentiated, or more frequently poorly differentiated—and a non-neuroendocrine population, both accounting for at least 30% of the whole tumor mass. Several studies recently focused on the key genetic and epigenetic changes underlying MiNENs to better understand how they develop, and explore biological similarities among the two components and their pure counterparts. However, their molecular landscape still remains poorly understood. NGS may represent a useful tool to study this orphan disease by detecting the main genetic alterations and possible therapeutic targets. NGS analysis on tissue and/or blood samples through the Foundation One (F1) platform was performed on consecutive samples collected from four patients diagnosed with MiNENs of the gastroenteric tract. Several genetic alterations were shared among samples from the same patients, thus suggesting a common origin between them, although morphology sometimes changed at histopathological evaluation. Common molecular alterations among samples from different patients that had not been previously described to our knowledge were also detected. Finally, it is of the utmost importance to clarify if the maintenance of the 30% cut-off is still essential in defining MiNENs and really manages to include all of the mixed neoplasms.

## 1. Introduction

Mixed neuroendocrine non-neuroendocrine neoplasms (MiNENs) represent a rare diagnosis of the gastrointestinal tract [1]. They refer to heterogenous neoplasms which may arise virtually in all organs, constituted of at least two distinct morphologically recognizable neoplastic components: a neuroendocrine population—either well-differentiated, or more frequently poorly differentiated—and a non-neuroendocrine population, both accounting for at least 30% of the whole tumor mass [2,3,4,5,6,7]. Compared to the previous definition of mixed adeno-neuroendocrine carcinomas (MANECs), the term MiNENs better represents the wide spectrum of all the possible combinations, as well as the variability of differentiation and morphology [3,4,7,8].

The pathogenesis of MiNENs is still a topic of open debate. The main question is whether MiNEN is arising from two separate clones (collision theory), or from a common multipotent progenitor stem cell with bidirectional differentiation (common precursor theory), or finally from a single non-neuroendocrine clone, whose neuroendocrine differentiation is the result of the progressive accumulation of genetic alterations and aberrations [5,6] (Figure 1). Although the concept of MINENs implies that the two components are clonally related, there are cases in which independent neoplasms coexist at the same site (collision theory), thus suggesting the possibility of a polyclonal origin for at least a subtype of MiNENs [6].

Due to the rarity of the diagnosis, the low quality of published data, and the plethora of different names used to address the same entity, limited data are available on the pathological features of this disease, as well as the most proper treatments.

Several studies have recently tried to identify the key genetic and epigenetic changes underlying MiNENs [5,9,10,11,12,13,14]: in a recent systematic review, Frizziero et al. offered a comprehensive overview of the main genetic alterations that have been reported to date in these tumors, regardless of their site of origin [5].

Regarding the best treatment choice, in the absence of data from clinical trials, the management of MiNENs depends on which of the two components is preponderant and/or shows the most aggressive histology. If the neuroendocrine component is poorly differentiated, MiNENs are generally treated according to the standard of care used for their pure NEC (neuroendocrine carcinoma) counterpart. Alternatively, when the non-neuroendocrine component is the most represented one and presents more aggressively, some clinicians choose to apply the standard of care for epithelial tumors from the same site of origin. However, both approaches are not supported by evidence from prospective randomized trials [5,15].

## 2. Materials and Methods

Foundation One (F1) testing was performed on consecutive histological samples and/or on blood samples collected from 4 patients diagnosed with MiNENs of the gastrointestinal tract. F1 test is a target specific NGS-based device for tissue and liquid biopsy of FoundationMedicine able to detect 324 molecular alterations: substitutions and indels, CNAs, selected genomic rearrangements, and genomic signatures—including tumor fraction (TF), blood tumor mutational burden (bTMB), and MSI-H status [16].

All the patients provided written informed consent for an institutional review board-approved protocol for collection of plasma and tumor DNA profiling.

## 3. Case Presentation

We report four cases of MiNENs of the gastrointestinal tract treated in our institution.

### 3.1. Case 1

A 74-year-old female was admitted to hospital in November 2016 with intestinal obstruction symptoms and emergency ileal resection with concomitant debulking surgery of an abdominal omental mass was performed. Histopathological diagnosis was consistent with NEC of unknown primary origin, made up of two components: large-size cells (20% of the whole neoplasm, Ki 67: 50%), and small-size cells (80% of the whole mass, Ki67: 90%). Five out of seven nodes showed metastatic involvement by a neoplastic neuroendocrine population. At immunohistochemical staining, both the populations were positive for chromogranin A and CD56. Synaptophysin staining was not available (Table 1).

A post-operatory CT scan showed multiple pathologic nodes and the presence of residual tumor tissue on the abdominal anterior wall. Chemotherapy with cisplatin g1 plus etoposide g1–3 q21 was administered for six cycles from January 2017 to May 2017. A partial response was achieved and maintained until June 2018.

In June 2018, for bowel obstruction signs, emergency colorectal resection and debulking surgery were performed. Histopathological examination showed neoplastic cells arranged in glands in tunica submucosa, muscularis, and subserosa without infiltrating serosa and mucosa. These findings were consistent with colorectal adenocarcinoma metastases localized in submucosa and tunica muscolaris. At immunostaining, CK19, CK7, and CDX2 were positive; while CD56, chromogranin, and synaptophysin were negative (Table 1).

Despite the absence of a neuroendocrine component at morphology and immunohistochemical staining, we hypothesized that the second neoplasm could be the expression of adenocarcinomatous counterpart selection of a previous high-grade MiNEN. At histological revision of the first sample, a small focus of epithelial cells showing high-grade dysplasia was present.

F1 analysis performed on both surgical samples described several similarities: the same tumor mutational burden, MS status, and TP53 mutation (*TP53 R273H*) were found as well as some common VUS alterations (*ERBB3*-L1177I-; *JAK3*-I63V-; *MED12*-L1348F-; *SDHC*-L14F-). Altogether, these findings may support the hypothesis of a common monoclonal origin of the two neoplasm (Table 2).

Based on a predominant expression of the adenocarcinomatous counterpart, the patient was administered a second-line therapy with FOLFOX from July 2018 to September 2019. Stable disease was the best response. For nodal and peritoneal progression, in March 2020 she underwent a third-line therapy with FOLFIRI plus ERBITUX. In March 2021, further peritoneal node enlargement was detected at CT scan and FOLFOX plus bevacizumab was started.

### 3.2. Case 2

A 50-year-old female patient was admitted to hospital in September 2018 for severe abdominal pain. A CT scan detected a mass in left colon with multiple liver metastases.

Histological examination performed on the colonic biopsy was consistent with a NEC characterized by cells with scant eosinophilic cytoplasm, and finely granular chromatin which were arranged in the lamina propria in an organoid growth pattern. Ki67 index was >55%. Immunohistochemical assays were not performed due to the scarcity of the tumor sample (Table 1). Liver biopsy was not feasible as the fast worsening of her clinical conditions and the onset of hepatic failure required salvage chemotherapy.

Carboplatin g1 plus etoposide g1–3 q21 was rapidly started. A CT scan, after four cycles, showed a partial response. Treatment was continued for a further four cycles. A CT scan performed in March 2019 was consistent with disease stability.

At colonoscopy, the tumoral mass was dramatically reduced and biopsies were consistent with colorectal adenocarcinoma G2. Revision of the first colonic samples by the same pathologists revealed the presence of colorectal high-grade dysplasia along with cells characterized by scant eosinophilic cytoplasm arranged in the lamina propria in an organoid growth pattern. As already mentioned above, immunohistochemistry was not performed due to the scarcity of the material (Table 1).

Subsequent laparoscopy showed the presence of several metastatic peritoneal implants that had not been detected by CT scan: hepatic resection of a single metastasis with peritoneal sampling were executed to better characterize the disease. At histological examination, they all showed a poorly differentiated adenocarcinoma (KI 67: 95%). At immunohistochemical analysis, chromogranin and synaptophysin were negative, while CK20 and CDX2 were positive (Table 1).

In April 2019, a second-line therapy with FOLFIRI+ bevacizumab was started. After 1 month, symptoms of neurological involvement occurred: cerebellar lesions were detected at brain MRI and soon after the patient died.

Altogether, the rapid onset as well as the site of occurrence, may suggest the neuroendocrine nature of these lesions, given the possibility of the two counterparts to progress and metastasize together but also separately.

F1 analysis on the first histologic samples collected in September 2018 identified the following mutations: *KRAS G12V*; *FBXW7 W673*; *PTEN N276K*; *APC D1486fs*21*0; and *TP53 R175H.* RICTOR amplification was detected and considered equivocal. The sample was MS stable and showed an intermediate tumor mutational burden (TMB) (13 Muts/Mb) (Table 2) F1 analysis performed on the hepatic sample collected in March 2019 revealed many similarities, despite the morphology having changed. The same alterations in *FBXW7* (*W673*), *KRAS* (*G12V*), *PTEN* (*N276K*), *APC* (*D1486fs*21*), and *TP53* (*R175H*) genes were described suggesting a common clonal origin among the two neoplasms and the early onset of these events. MS stability and intermediate TMB (11 Muts/Mb) were also reported, as in the primary tumor. Several VUS were also shared among the two samples (Table 2).


healthcare-10-00708-t001_Table 1Table 1Overview of the morphological and immunohistochemical features of all the tumor samples evaluated.
SampleMorphologyImmunohistochemistrySampleMorphologyImmunohistochemistry
**Case 1**
surgicalTwo neoplastic neuroendocrine components: large-size cells (20% of the whole neoplasm, Ki 67%: 50%), and small-size cells (80% of the whole mass, Ki 67: 90%).Five out of seven nodes showed metastic involvement by the neoplastic neuroendocrine population**Revision**:Presence of focal epithelial high-grade dysplasia Chromogranin A and CD56 positive in both the populations.SurgicalCells arranged in glands were described in tunica submucosa, muscularis, and subserosa without involving serosa and mucosa.Positivity for CK19, CK7, and CDX2;CD56, chromogranin, and synaptophysin all negative.
**Case 2**
biopticCells characterized by scant eosinophilic cytoplasm, finely granular chromatin, arranged in the lamina propria in an organoid growth pattern. Ki 67 > 55%. **Revision**:Colorectal high-grade dysplasia along with cells characterized by scant eosinophilic cytoplasm, finely granular chromatin arranged in the lamina propria in an organoid growth pattern.Not performed due to the scarcity of tumor sample.Surgical liverPoorly differentiated neoplasia(Ki 67: 95%).Positivity for CDX2 and CK20; chromogranin and synaptophysin negative.
**Case 3**
surgicalTwo neoplastic components: a colorectal adenocarcinoma G2 occupying 60% of the whole neoplasm and a neuroendocrine carcinoma (40% of the tumor) composed of small cells.Several necrotic areas were presentTumor deposits of neuroendocrine cells were detected. Tunica serosa was infiltrated and ulcerated by both the populations as well as one node out of 11.Focal positivity for CK20, synaptophysin, and CDX2.



**Case 4**
surgicalTwo populations, each occupying 50% of the whole mass: mucinous adenocarcinoma with signet ring cells and neuroendocrine carcinoma. Tumor cells, diffusely occupied submucosa and muscolaris tunica with just focal mucosal involvement.One out of seven nodes showed neoplastic involvement by both the populations.High and diffuse positivity for CK19 and CDX2; low but diffuse positivity for CK20; CK-7 negative. Neuroendocrine markers were just focally positive (dispersed).






healthcare-10-00708-t002_Table 2Table 2Overview of the main alterations detected in the samples examined (surgical, bioptic, blood), at least two for each case except for Case 4. Comparison among samples collected at different times in the same patient was made in order to better understand disease evolution and the behavior of its counterparts: shared alterations were highlighted in red, while new ones were reported.
First NGSMSTMBVUSSampleDiagnosisSecond NGSMSTMBVUSSampleDiagnosis
**Case 1**
***AKT2*** amplification***AXL*** amplification***CCNE1*** amplification***MYCL1*** amplification ***MAP2K4*** loss***RB1*** loss exons 1–2TP53 R273H
MSSLow (4 Muts/Mb)**Common**:**ERBB3**L1177I**JAK3**E113K and I63V**MED12**L1348F**SDHC**L14F**Not shared**:**CD22**amplification**CD79A**amplification**CEBPA**amplification**CIC**amplification**DOT1L**T1460K**TET2**F1597LSurgical
**NEC of unknown origin**

**Revision:**

**Presence of focal epithelial high-grade dysplasia**
***TP53*** R273H**DNMT3A**R676W—subclonalMSSLow(1 Muts/Mb)***Common***:***ERBB3***L1177I***JAK3***I63V***MED12***L1348F***SDHC***L14FSurgicalMetastases of colon adenocarcinoma 
**Case 2**

***KRAS***G12V

***FBXW7***W673*

***PTEN*** N276K

***RICTOR*** amplification—equivocal

***APC*** D1486fs*21

***TP53*** R175H
MSSIntermediate (13 Muts/Mb)**Common**:***BCORL1***N1473I***DOT1L***P1354L***FANCC***D306E***KDR***A379V***MAP3K1***S939C***MED12***Q2119_Q2120insHQQQ***PIK3C2G***F89Y***SMAD4***Y353H***RICTOR***I253V**Not shared**:*NTRK3*H521DbiopticNEC of colonRevision:Presence of high-grade dysplasia in a context of neuronendocrine cells

*
**KRAS**
*


G12V

***FBXW7*** W673*

***PTEN*** N276K

***APC*** D1486fs*21


**
*TP53*
**


R175H
MSSIntermediate (12 Muts/Mb)**Common**:***BCORL1***N1473I***DOT1L***P1354L***FANCC***D306E***KDR***A379V***MAP3K1***S939C***MED12***Q2119_Q2120insHQQQ***PIK3C2G***F89Y***SMAD4***Y353H***RICTOR***I253V + amplification**Not shared**:*FGF10*amplification*SDHA*amplification**RICTOR**amplificationSurgical liverColorectaladenocarcinoma 
**Case 3**
**KRAS** G12D **APC** R876*I1557fs*1**FAM123B** R601***TP53** R196*C242F**Not shared**:**SMAD4** P356L **MYC-N** amplificationMSS8 Muts/Mb**Common****ATR**D331G**BRIP1**R106C**CD79A**T75M**HGF**W329L**LTK**V113L**Not shared**:**DIS3**amplification**FGF14**amplification**FGF6**G187R**IRS2**amplificationSurgicalMINEC(colorectal adenocarcinoma G2+ NEC)**KRAS** G12D **APC** R876*, I1557fs*1**FAM123B** R601***TP53** R196***Not shared**:**ASXL1** R1415***DNMT3A**W440*, N552fs*99**ERBB2** S310Y**STAG2** splice site 2924+2T>GMSS8 Muts/Mb**Common**:**ATR**D331G**BRIP1**R106C**CD79A**T75M**HGF**W329L**LTK**V113L**Not shared**:**DNMT3A**I369F**EPHB1**P844S**MAP3K13**R474Q**ATM**F2485G**PDGFRA**H425R**RICTOR**V120fs*2BloodColorectal MINEC
**Case 4**
**NF1**loss exons 1–36**CTNNB1** D32N**TP53** K382fs*40**IRS2** amplificationMSS8 Muts/Mb**CDH1**D587N**DIS3**K923N**ERBB4**M887I**FGF12**G82***NTRK1**G714S**NTRK3**I533L**PIK3C2B**D777A**PIK3C2G**I232T**STK11**S240L**TEK**I591VSurgicalMINEC(mucinous adenocarcinoma + NEC) of unknown primary origin(gastric, colon?)
**Not available**








### 3.3. Case 3

In December 2020, a 71-year-old male was admitted to hospital for abdominal pain and bowel obstruction signs. The patient had a previous history of colorectal surgery for unspecified reasons. An emergency CT scan showed thickening of the anastomotic walls with perivisceral fat and ileal involvement. Right hemicolectomy with partial ileal resection and omentectomy was performed in urgency.

Histopathological evaluation was consistent with the diagnosis of mixed tumor made up of two components: a colorectal adenocarcinoma G2 (60% of the whole neoplasm) and a neuroendocrine carcinoma (40% of the mass) composed of small cells—focally positive for CK20, synaptophysin, and CDX2—with several necrotic areas (Figure 2A–E). Tumor deposits of neuroendocrine cells were detected. Tunica serosa was infiltrated and ulcerated by both the neoplastic components. One node out of 11 showed also metastatic involvement by both the neoplastic populations (Table 1).

All these findings were consistent with high-grade colorectal MiNEN.

Post-operatory CT scan performed in January 2021 revealed the presence of abdominal residual disease infiltrating pancreas, duodenum, and mesenteric vessels. Mesenteric peritoneal and nodal involvement was also described. A first-line therapy with platinum g1 plus etoposide g1–3 q21 was started.

F1 analysis on surgical sample showed alterations in *KRAS* (*G12D*) *APC* (*I1557fs*1*, *R876**), *SMAD4* (*P356L*), *TP53* (*R196**, *C242F**), and *FAM123B* (*R601**) genes. MYC-N amplification was also described. The tumor was MS stable and mutational burden was assessed at 8 Muts/Mb (Table 2).

CT scan after four cycles revealed peritoneal progression and F1 liquid analysis was performed.

The same mutations in *KRAS*, *APC*, *FAM123B*, and *TP53* genes were identified as well as the same TMB (8 Muts/Mb). However, other alterations were detected only in blood: the genes involved were *ERBB2* (*S310Y*), *ASXL1* (*R1415**) *DNMT3A* (*W440**, *N552fs*99*:), and S*TAG2* (splice site 2924+2T>G). *MYC-N* amplification was not described anymore. Several VUS were also shared among the two samples (see Table 2 for details).

Given the peritoneal progression of disease, the patient started a second-line therapy with FOLFOX that is still ongoing. Partial response was achieved and is still maintained.

### 3.4. Case 4

In November 2020, a 50-year-old male was admitted to the hospital for abdominal pain, alteration in stool frequency, and vomiting. Clinical diagnosis of upper GI sub-occlusion was made and confirmed by urgency CT scan. EGD revealed huge dilatation of the stomach with pyloric stenosis. Biopsies were performed and confirmed the neoplastic nature of the lesion.

The patient underwent subtotal gastrectomy and lymphadenectomy. Histopathological examination was consistent with a mixed tumor composed of two populations, each occupying 50% of the whole mass: mucinous adenocarcinoma with signet ring cells and neuroendocrine carcinoma. Tumor cells, diffusely occupied submucosa and muscolaris tunica with just focal mucosal involvement (Figure 3A–C). At immunohistochemical staining, CK19 and CDX2 showed high and diffuse positivity, CK20 low but diffuse positivity, while CK-7 was negative. Neuroendocrine marker positivity was dispersed, but the percentage was not defined. One out of seven nodes showed neoplastic involvement by both the neoplastic populations (Table 1). CK20 positivity, CK7 negativity as well as the focal mucosal involvement could suggest the metastatic origin from lower GI tract rather than a gastric primitivity. However, at colonoscopy, no colonic lesions were detected.

Delayed post-operatory CT scan was performed in March 2021 for patient’s will. At abdominal evaluation, two sub-centimetric nodes were detected and were suggestive of peritoneal carcinomatosis.

F1 test on surgical sample showed *NF1* gene loss (exons 1–36), *CTNNB1 (D32N)* and *TP53 (K382fs*40*), gene mutations and finally IRS2 amplification. The tumor was MS stable and TMB was assessed at 8 Muts/Mb (Table 2). Moreover, several VUS were detected in the following genes: *CDH1 (D587N)*, *DIS3 (K923N)*, *ERBB4 (M887I)*, *FGF12 (G82*)*, *NTRK1 (G714S)*, *NTRK3 (I533L)*, *PIK3C2B (D777A)*, *PIK3C2G (I232T)*, *STK11(S240L)*, and *TEK (I591V)* (see Table 1). Although some interesting target genes were involved, these alterations have not been related to a pathogenetic potential. 

In May 2021, the patient started a first-line chemotherapy with cisplatin g1 + etoposide g1–3 q21. CT scan after 3 months showed disease stability and further three cycles were administered. 

In October 2021, after six cycles, CT and MRI were performed. As the response was maintained, the patient was addressed at follow-up.

## 4. Discussion

MiNENs represent a topic of open debate. Their rarity, as well as the absence of a common terminology, has led to underestimate their real incidence. The 2017 consensus may help to find a common path to follow. However, several questions are still open and require common evidence-based answers.

Although a comprehensive overview of the main molecular and genetic alterations was recently published [5], we compared consecutive samples of the same patient collected at baseline and at progression through F1. For the patient in Case 4, only one sample was analyzed as he did not experience disease progression under treatment.

Furthermore, being the threshold value of 30% arbitrarily assessed based on the first indications by Lewin, we questioned if it is really representative of this category and manages to include all the mixed neoplasms [7].

An extensive evaluation of tumor samples is mandatory. Surgical samples should always be preferred: the intrinsic limitation of bioptic samples in diagnosing MiNENs is a critical issue.

Frizziero et al. reported that initial biopsy was able to identify the presence of a mixed histology in only a third of cases [5]: the paucity of tissue, the heterogenous distribution of the components in the sample as well as the possible prevalence of only one of the two components in a particular site may explain these findings.

Biopsy specimens may be scantly representative of the whole mass, as happened in Case 2. In this patient, the first colonic biopsy was consistent with a pure NEC without an epithelial component, that was detected only at revision. This may explain the presence of adenocarcinoma in the subsequent colonic biopsy and laparoscopic samples. It is likely that an epithelial component was already present in the primary tumor but was not adequately represented in the bioptic sample.

Interestingly, in Case 1, even in the presence of a surgical sample, a limited epithelial counterpart in the context of neuroendocrine carcinoma was described only after revision.

According to the most recent guidelines both in Case 1 and in Case 2, we could not consider the tumors as MiNENs. However, a question may rise whether the universally accepted quantitative threshold is really adequate and representative for such a complex disease, being an arbitrary criterion, whose validity has not been demonstrated by systematic studies [7]. In the last few years, the hypothesis that even a minor component of high-grade NEN, irrespective of the quantitative threshold, can drive tumor prognosis has been gaining ground in clinical practice [7].

Interestingly, at NGS analysis, several genetic alterations were shared among the first and the second tissue samples both in Case 1 and Case 2: this event supports the hypothesis of a common monoclonal origin of the two neoplasms. It is likely that, under the pressure of neuroendocrine standard therapy, there was a selection of the adenocarcinomatous counterpart of a primitive high-grade MiNEN that may explain the subsequent histopathological findings.

Another intriguing topic to consider is the chance that both components may metastasize in an independent way [5,7]. For this reason, re-biopsy in case of relapse should be evaluated.

In Case 2, the laparoscopic samples were consistent with poorly differentiated adenocarcinoma. Furthermore, the onset of cerebral metastases which is not an early and common event in colorectal adenocarcinoma natural history (especially after only one cycle of therapy) may suggest a neuroendocrine phenotype, being neurotropism more frequent for such tumors. We hypothesized that, as soon as carboplatin plus etoposide administration was stopped, neuroendocrine counterpart progressively increased and new mutant aggressive clones selected by therapy could have migrated towards the central nervous system.

The comparison among F1 reports of consecutive samples collected in the same patients, revealed several shared pathogenic alterations as well as VUS. These findings supported the existence of a close relationship among the tumors.

Two out of four patients showed DNMT3A mutations. Interestingly, in their recent metanalysis, Frizziero et al. did not mention these alterations among the main molecular findings reported in MiNENs [6]. DNMT3A gene encodes DNA methyltransferase 3A, an enzyme involved in de novo methylation of DNA, whose role in cancer is still uncertain [17]. Alterations such as those seen here, may disrupt DNMT3A function or expression and cause impaired functioning. Rahman et described the overexpression of DNMT1, 3a, and 3b in several GEP-NENs as a common event and found that it was significantly higher in stage III and IV GEP NENs than in stage I or II, as well as in poorly differentiated neuroendocrine tumors compared to well-differentiated ones [18].

In Case 3, MYC-N amplification was described in the first tissue sample of high-grade colonic MiNEN. MYC-N amplifications have been reported in fewer than 1.0% of colorectal cancer [19], while Frizziero et al. mentioned them among the most frequent alterations found in MiNENs [5,11,14]. It is worth noting that c-MYC and SMARC4A have been considered as the potential mediators of the trans-differentiation process of the non-neuroendocrine component towards the more aggressive neuroendocrine counterpart, given the theory of the common intestinal ancestor [5].

In Case 4, a mutation in *CTNNB1* gene was described. These alterations have been reported in some neuroendocrine tumors analyzed in COSMIC, including carcinoid tumors of the duodenum (44%), stomach (21%), and large intestine (12%) (COSMIC, Jan 2021) [20]. Exon 3 mutations of this gene are responsible for increased beta-catenin protein stability and activation of the WNT pathway [21]. Interestingly, high expression of beta-catenin and integrity of E-cadherin–catenin complex (as happened in this case) have been associated with better survival rates in patients with gastro-enteropancreatic NET [22].

Amplification in *IRS2* was also identified in this patient. IRS2 is a key mediator of IR signaling as well as of IGF-1R pathway. I*RS2* amplifications have been described in several pure neuroendocrine carcinomas of different sites—such as small cell NEC (SCNEC) of the cervix, small cell lung cancer (SCLC), and pulmonary LCNEC [23,24,25,26]. Interestingly, 2–5% of SCLC patients harbor these alterations that may represent a predictive biomarker of response to ceritinib [24,25].

## 5. Conclusions

The question about threshold values is still open and deserves proper answers to avoid underestimation as well as underdiagnosis of MiNENs.

NGS analysis performed on consecutive samples detected several shared alterations which supported the hypothesis of a common origin between neoplasms occurred in the same patient, even in the presence of a different morphology, thus suggesting a potential role for genomic profiling in this complex and still poorly explored scenario.

Further studies are needed to better understand such a complex new-born entity and define which molecular alterations are exclusive of these neoplasms, which of them are shared by their pure counterparts, and how they interplay to give rise to this mixed fascinating entity.

## Figures and Tables

**Figure 1 healthcare-10-00708-f001:**
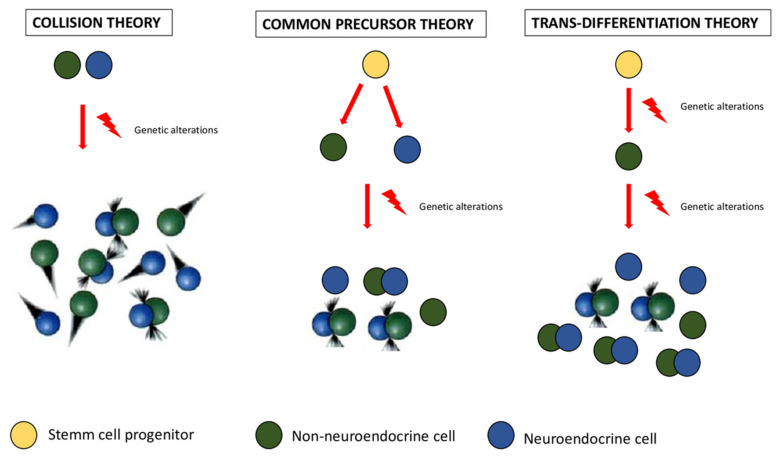
A comprehensive representation of the three main theories about MiNENs’ origin proposed to date. From the left to the right, collision theory, common precursor theory, and trans-differentiation theory are illustrated respectively. As mentioned in the text, the first one suggests that neuroendocrine and non-neuroendocrine components arise independently from distinct precursor cells, the second one postulates a common origin from a pluripotent stem cell progenitor, which differentiates towards both components; finally, the third one assumes that neuroendocrine differentiation is the result of progressive accumulation of genetic alterations in an initially non-neuroendocrine cell phenotype.

**Figure 2 healthcare-10-00708-f002:**
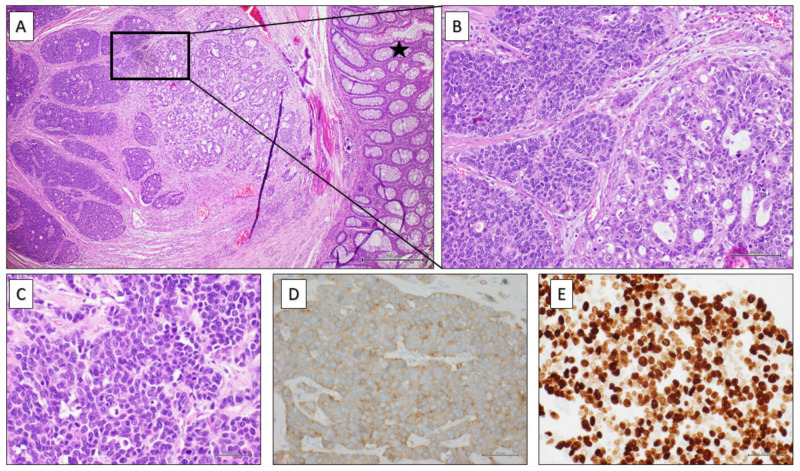
(**A**) Colonic segment partially covered by normal mucosa (black star) showing infiltration, on the left side of the image, by a carcinoma with a double component: neuroendocrine and adenocarcinomatous one (H&E 4× magnification). (**B**) A detail of image (**A**) showing at higher magnification the neuroendocrine population (on the left side) arranged in pseudo-rosettes or trabeculae, composed of medium-sized cells with irregular nuclei and granular chromatin and the adenocarcinomatous component (on the right side) with evident glandular structures made up of atypical cylindrical cells (H&E 20× magnification). (**C**) High mitotic index typical of neuroendocrine carcinomas (H&E 40× magnification). (**D**) Strong positivity for synaptophysin at immunohistochemistry (40× magnification). (**E**) High proliferative index (Ki67) at immunohistochemical assay (40× magnification).

**Figure 3 healthcare-10-00708-f003:**
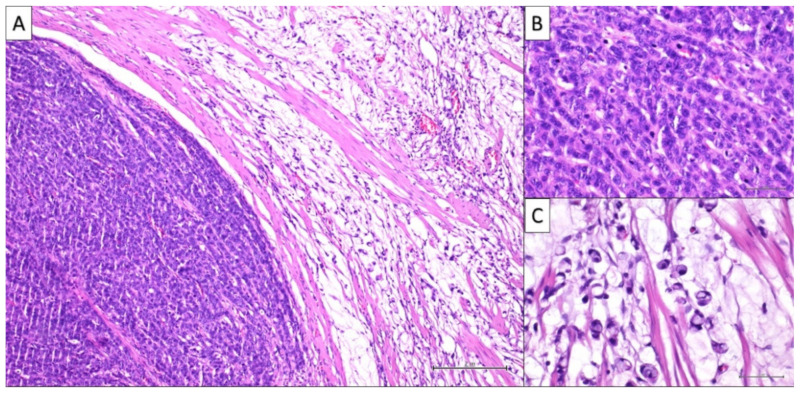
(**A**) Histologically, a carcinoma with a double component is observed: a neuroendocrine population (on the left side), and a poorly differentiated adenocarcinoma (on the right side) are clearly recognizable (H&E 10× magnification). (**B**) A detail of the neuroendocrine component at higher magnification showing a neoplastic population with cord architecture, medium-sized elements with scant cytoplasm, atypical nuclei, and numerous mitotic figures (H&E 20× magnification). (**C**) A detail of the adenocarcinomatous counterpart at higher magnification showing abundant mucus lakes with ‘floating’ scattered signet ring cells (H&E 20× magnification).

## Data Availability

Not applicable.

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
