# Peer review of "Mixed Neuroendocrine Non-Neuroendocrine Neoplasms of the Gastrointestinal Tract: A Case Series"

_healthcare, 2022, doi:10.3390/healthcare10040708_

Round 1

Reviewer 1 Report

It is a very interesting and well structured topic. The cases are few, given the rarity of the neoplasm, I am not convinced only by the negativity of the neuroendocrine markers. In addition to chromogranin A and CD56, the authors also tried a staining for synaptophysin?

Neuroendocrine and nonneuroendocrine components are confirmed with immunohistochemistry (IHC) (WHO 2019)

The authors must clarify the morphological and immunohistochemical characteristics of all the pathological characteristics in the tumor samples evaluated and also in the lymph nodes where present.

Case 2: In this case "The review of the first colon samples by the same pathologists revealed the presence of high-grade colorectal dysplasia along with cells characterized by poor eosinophilic cytoplasm, finely granular chromatin that were arranged in the lamina propria according to a pattern of organoid growth ". Have immunophenotypic stains been performed for neuroendocrine markers?

Case 3: “Tumor deposits of neuroendocrine cells have been detected. The serous tunic was infiltrated and ulcerated. One node in 11 showed metastatic involvement “clarify the claim. Was the metastatic involvement of neuroendocrine tumor or colorectal adenocarcinoma?

The same kind of consideration in the case 4

Author Response

PLEASE SEE THE ATTACHEMENT 

Reviewer 2 Report

The paper raises exciting issues for readers.

The authors might be interested in improving their report according to the following comments and suggestions.

According to the current WHO-DSTs (2019):

1) MiNEM are considered mixed epithelial neoplasms in which a neuroendocrine (mostly poorly differentiated: SCNE or LCNEC) component is combined with a non-neuroendocrine component, each of which is morphologically and immunohistochemically (e.g., synaptophysin, chromogranin A, strongly recommended for SCNEC or LNEC) recognizable as a discrete component and constitutes (by arbitrary convention) ≥ 30% of the neoplasm (the two components are presumed to be clonally related) ;

2) An important consideration is finding focal (<30%) SNEC associated with a non-neuroendocrine neoplasm. Because of the clear clinical significance of SNEC, even minor components should be mentioned in the diagnosis.

 3) MiNEM is regarded as a conceptual category of neoplasms rather than a specific diagnosis.

The author's state:

 "Due to the rarity of the diagnosis, the low quality of published data, the plethora of different names used to address the same entity, limited data are available on the pathological features of this disease as well as the most proper treatments."

"In these cases an extensive evaluation of tumour samples is mandatory."

"NGS analysis on tissue and or blood samples through Foundation One (F1) platform was performed on consecutive samples collected from each of the four cases of patients diagnosed with MiNENs (this diagnosis is stated only in two cases in Table 1).

"The NGS F1 findings may support the hypothesis of a common monoclonal origin (case 1), suggesting a common clonal origin and the early onset of these events (case 2), although morphology sometimes changed at histopathological evaluation."

The paper should improve according to the following suggestions:

a) For consistency of data report, the authors must systematically clarify in the paper (a new summary Table would be warranted) the morphological and immunohistochemical features of all the pathology descriptions in tumor samples evaluated during the follow-up of each patient, according to the current WHO guidelines (see points 1 and 2).

b) Case 1: specify the morphology and immunohistochemistry of "the 5 out of 7 nodes showed neoplastic involvement" (ileal resection with concomitant debulking surgery of an abdominal omental mass); clarify the lymph node status in the "colorectal resection and debulking surgery."

c) Case 2: clarify the statement "Revision of the first colonic samples by the same pathologists revealed the presence of colorectal high-grade dysplasia along with cells characterized by scant eosinophilic cytoplasm, finely granular chromatin which were arranged in the lamina propria in an organoid growth pattern" (see also point a).

d) Case 3: clarify the statement, "The patient had a previous history of colorectal surgery for unspecified reasons. thickening of the anastomotic walls with perivisceral fat and ileal involvement. Right hemicolectomy with partial ileal resection and omentectomy was performed"; i.e., is it not common/good practice to ask/request revision of pathology reports/material in such cases?

e) Case 4: clarify the statement "Neuroendocrine markers were just focally positive" i.e., dispersed or grouped cells/percentage of cells (see also point a)?.

f) Clarify, considering point 3 and that there is no description stating that the two components of the tumors were systematically analyzed separately. How can one support their common clonal origin hypothesis and interrogate intra-tumor heterogeneity and clonal evolution?

g) Comment briefly in the discussion or introduction (concerning Fig.1) the possibility of a polyclonal origin for at least a subtype of MiNENs (Ref.7).

Author Response

Napoli, Italy;

02 April 2022

Dr. Rahman Shiri Website SciProfiles, Editor-in-Chief; Ms. Ivana Prdic, BSc, Assistant Editor

Healthcare, MDPI

Dear all,

Enclosed please find the revised version of the manuscript “Mixed neuroendocrine non-neuroendocrine neoplasms of the gastrointestinal tract: a case series.”.

We revised thoroughly the paper according to the comments from the Reviewer and Editors. The suggestions proposed by the Reviewer and the Editors have been useful to improve the quality of the work and reinforce the message that the Authors wish to deliver. We hope that this manuscript is now appropriate for publication by Healthcare, consistently with its aim to improve knowledge and quality of cancer care worldwide.

This manuscript has not been published and is not under consideration for publication elsewhere. Additionally, all the authors have approved the contents of this paper and have agreed to the Healthcare’s submission policies.

The following is a list of point-by-point answers to the comments.

We hope you find our manuscript suitable for publication.

Thank you for your consideration of this manuscript.

Sincerely,

Stefania Napolitano, MD, PhD;

Department of Precision Medicine,

Università della Campania “Luigi Vanvitelli”,

  1. Pansini 5, 80131, Napoli (Italy); Mail: [email protected]; Phone: 003908156664192

Reviewer 2

The paper raises exciting issues for readers.

The authors might be interested in improving their report according to the following comments and suggestions.

According to the current WHO-DSTs (2019):

1) MiNEM are considered mixed epithelial neoplasms in which a neuroendocrine (mostly poorly differentiated: SCNE or LCNEC) component is combined with a non-neuroendocrine component, each of which is morphologically and immunohistochemically (e.g., synaptophysin, chromogranin A, strongly recommended for SCNEC or LNEC) recognizable as a discrete component and constitutes (by arbitrary convention) ≥ 30% of the neoplasm (the two components are presumed to be clonally related)

2) An important consideration is finding focal (<30%) SNEC associated with a non-neuroendocrine neoplasm. Because of the clear clinical significance of SNEC, even minor components should be mentioned in the diagnosis.

 3) MiNEM is regarded as a conceptual category of neoplasms rather than a specific diagnosis.

The author's state:

 "Due to the rarity of the diagnosis, the low quality of published data, the plethora of different names used to address the same entity, limited data are available on the pathological features of this disease as well as the most proper treatments."

"In these cases, an extensive evaluation of tumour samples is mandatory."

"NGS analysis on tissue and or blood samples through Foundation One (F1) platform was performed on consecutive samples collected from each of the four cases of patients diagnosed with MiNENs (this diagnosis is stated only in two cases in Table 1).

"The NGS F1 findings may support the hypothesis of a common monoclonal origin (case 1), suggesting a common clonal origin and the early onset of these events (case 2), although morphology sometimes changed at histopathological evaluation."

The paper should improve according to the following suggestions:

  1. a) For consistency of data report, the authors must systematically clarify in the paper (a new summary Table would be warranted) the morphological and immunohistochemical features of all the pathology descriptions in tumor samples evaluated during the follow-up of each patient, according to the current WHO guidelines (see points 1 and 2).

Answer: We thank referee for the precious suggestion. Morphological and immunohistochemical features were described about the first sample of case 1.  Unfortunately, for second neoplasm, Immunohistochemical assays were not performed due to the scarcity of tumour sample. Regarding case 2-4 all featured described in histological diagnosis have been reported in the manuscript.  A new summary table with all morphological and immunohistochemical features have been added to the paper, according to referee suggestions. 

  1. b) Case 1: specify the morphology and immunohistochemistry of "the 5 out of 7 nodes showed neoplastic involvement"(ileal resection with concomitant debulking surgery of an abdominal omental mass);clarify the lymph node status in the "colorectal resection and debulking surgery."

Answer: We thank referee for this observation. The nodes involved showed infiltration by the same population described in the omental mass: a neoplastic population made of small and large size cells was found which was positive for CD56 and chromogranin A at immunostaining.

  1. c) Case 2: clarify the statement "Revision of the first colonic samples by the same pathologists revealed the presence of colorectal high-grade dysplasia along with cells characterized by scant eosinophilic cytoplasm, finely granular chromatin which were arranged in the lamina propria in an organoid growth pattern" (see also point a).

Answer: We thank referee for this evaluable observation. High-grade epithelial dysplasia was found. However, this condition is not considered for the diagnosis of MINEN. We just reported this finding and questioned whether this component could explain the subsequent finding of pure adenocarcinoma without neuroendocrine component.

  1. d) Case 3: clarify the statement, "The patient had a previous history of colorectal surgery for unspecified reasons. thickening of the anastomotic walls with perivisceral fat and ileal involvement. Right hemicolectomy with partial ileal resection and omentectomy was performed"; i.e., is it not common/good practice to ask/request revision of pathology reports/material in such cases?

Answer: We thank referee for this observation. Yes, it is a common practice of our Department. However, although we requested pathological samples, they were not available. We apologize for missing these data.

  1. e) Case 4: clarify the statement "Neuroendocrine markers were just focally positive" i.e., dispersed or grouped cells/percentage of cells (see also point a)?

Answer: We thank referee for this observation. At morphological evaluation, the neoplastic neuroendocrine population represented 50% of the whole mass. Neuroendocrine markers positivity was dispersed but the percentage was not defined. We modify this sentence according to referee suggestion.

  1. f) Clarify, considering point 3 and that there is no description stating that the two components of the tumors were systematically analyzed separately. How can one support their common clonal origin hypothesis and interrogate intra-tumor heterogeneity and clonal evolution?

Answer: We thank referee for this precious observation.  We are aware that the two components were not micro-dissected and analyzed separately and that this represents a limit of our study. However, the common monoclonal origin was hypothesized among the primary tumors and the second neoplasms in case 1 and in case 2, because similar genetic alterations were found despite the different morphology. We were not referring to the theory of the common monoclonal origin of both components in MiNEN pathogenesis.

  1. g) Comment briefly in the discussion or introduction (concerning Fig.1) the possibility of a polyclonal origin for at least a subtype of MiNENs (Ref.7).

Answer: We thank referee for this observation. The possibility of a polyclonal origin has been discussed in the introduction part, according to referee suggestion.

Please, see the attachment too

Round 2

Reviewer 2 Report

It seems that the revised manuscript may be o interest to readers.

This manuscript is a resubmission of an earlier submission. The following is a list of the peer review reports and author responses from that submission.